# The Changes in the p53 Protein across the Animal Kingdom Point to Its Involvement in Longevity

**DOI:** 10.3390/ijms22168512

**Published:** 2021-08-07

**Authors:** Martin Bartas, Václav Brázda, Adriana Volná, Jiří Červeň, Petr Pečinka, Joanna E. Zawacka-Pankau

**Affiliations:** 1Department of Biology and Ecology, Faculty of Science, University of Ostrava, 71000 Ostrava, Czech Republic; martin.bartas@osu.cz (M.B.); jiri.cerven@osu.cz (J.Č.); petr.pecinka@osu.cz (P.P.); 2Institute of Biophysics of the Czech Academy of Sciences, 61265 Brno, Czech Republic; vaclav@ibp.cz; 3Department of Physics, Faculty of Science, University of Ostrava, Chittussiho 10, 71000 Ostrava, Czech Republic; adriana.volna@osu.cz; 4Faculty of Chemistry, University of Warsaw, Pasteura 1, 02-093 Warsaw, Poland; 5Center for Hematology and Regenerative Medicine, Department of Medicine, Huddinge, Karolinska Institute, SE 171-74 Stockholm, Sweden

**Keywords:** p53, aging, longevity, comparative analysis, protein sequence

## Abstract

Recently, the quest for the mythical fountain of youth has produced extensive research programs that aim to extend the healthy lifespan of humans. Despite advances in our understanding of the aging process, the surprisingly extended lifespan and cancer resistance of some animal species remain unexplained. The p53 protein plays a crucial role in tumor suppression, tissue homeostasis, and aging. Long-lived, cancer-free African elephants have 20 copies of the *TP*53 gene, including 19 retrogenes (38 alleles), which are partially active, whereas humans possess only one copy of *TP*53 and have an estimated cancer mortality rate of 11–25%. The mechanism through which p53 contributes to the resolution of the Peto’s paradox in Animalia remains vague. Thus, in this work, we took advantage of the available datasets and inspected the p53 amino acid sequence of phylogenetically related organisms that show variations in their lifespans. We discovered new correlations between specific amino acid deviations in p53 and the lifespans across different animal species. We found that species with extended lifespans have certain characteristic amino acid substitutions in the p53 DNA-binding domain that alter its function, as depicted from the Phenotypic Annotation of p53 Mutations, using the PROVEAN tool or SWISS-MODEL workflow. In addition, the loop 2 region of the human p53 DNA-binding domain was identified as the longest region that was associated with longevity. The 3D model revealed variations in the loop 2 structure in long-lived species when compared with human p53. Our findings show a direct association between specific amino acid residues in p53 protein, changes in p53 functionality, and the extended animal lifespan, and further highlight the importance of p53 protein in aging.

## 1. Introduction

The promise of eternal life has inspired research into this topic across many civilizations and through the millennia, dating back to Herodotus and his writings 2500 years ago. Although the average human lifespan is increasing, our health span appears to be lagging. Several studies argue that the human lifespan is physiologically and genetically limited [1], yet recent contributions have proposed a future with a potentially unlimited increase in human lifespan [2]. The demographical data show that the death risk increases exponentially up to about age 80, then decelerates and plateaus after age 105 [3]. There are two major theories of aging, senescence theory and programmed theory of aging [4]. The senescence theory converges on the accumulation of cellular damage that cannot be repaired, leading first to permanent cell cycle arrest and, in the end, the loss of organismal fitness. The free radical theory can be classified as a subtype of senescence theory and postulates that organisms age because of the accumulation of the damage inflicted by reactive oxygen species [5,6]. There is also a common agreement that the preservation in the fidelity of the DNA repair process involving the p53 pathway favors longevity [7]. The programmed theory of aging states that aging is tightly controlled and includes the Hayflick limit theory and the central aging clock theory. At the molecular level, biological aging is a complex process that involves genetic factors, mitochondria damage mechanisms, cellular senescence, proteostasis and autophagy, telomere attrition, epigenetics, inflammation, and metabolic switches. Thus, the lifespan is a multi-nodal characteristic [8]. To date, several factors have been found to play important roles in human aging, including mammalian target of rapamycin (mTOR), 5′ AMP-activated protein kinase (AMPK), sirtuin 1 (SIRT1), peroxisome proliferator-activated receptor gamma coactivator 1-alpha (PGC-1α), apolipoprotein E (APOE), lipoprotein (A) (LPA), CDKN2B antisense RNA 1 (CDKN2B-AS1), and p53. Among those, p53 emerges as a central node, linking several pathways together. The p53 is a tumor suppressor that is coded by the most often mutated gene in human cancers [9,10,11,12], and the loss of wild-type p53 function is associated with fatal outcomes in cancer patients. p53 is a critical sensor of cellular stress and thus, the dictator of cell fates. Depending on the types of stress, which include DNA damage, oncogene activation, nutrient deprivation, reactive oxygen species accumulation, and telomere shortening, p53 either (1) transiently stops cell proliferation, initiates the DNA repair machinery, and induces cell death when the damage cannot be repaired, or (2) pushes cells to replicative senescence, which is a permanent proliferation arrest.

Given the high cancer susceptibility in humans and the role of p53 in regulating cell fate, p53 is regarded as the key regulator of humans’ healthy lifespan [13,14]. When we consider the “lifespan” of tumor cells, it is apparent that cancer cells often gain new functions, including “immortality,” which is at least partially attributed to the inactivating mutations in the *TP*53 gene and/or in its regulatory pathways [15]. As reviewed by Stiewe and Haran [16], cancer-associated mutations alter p53 in three ways: they promote the loss of wild-type (wt) p53 DNA binding, trigger dominant-negative inhibition of wtp53 by the mutant p53 in the monoallelic mutation setting, or induce the gain of new functions by mutant p53 through new protein–protein–DNA interactions. The loss of binding to the canonical target sequence by mutant p53 can be partial or complete. Different mutant p53 proteins show a variable degree of loss of the DNA-binding capacity. This results in attenuated or target-selective DNA-binding patterns [16]. Multiple functions of p53 have been described and extensively reviewed [17,18,19]. For example, the p53 protein plays roles in metabolism [20], cell cycle arrest [21,22], apoptosis [23], ferroptosis, angiogenesis [24], DNA repair [25], embryonic development, and cell senescence [18,26]. In the majority of cellular processes, p53 functions as a transcription factor and recognizes and binds to multiple target genes through a recognition sequence (5′-PuPuPuC(A/T)(T/A)GPyPyPy-3′) [27,28,29]. Owing to its crucial role in protection against the accumulation of DNA damage, p53 is called “the guardian of the genome” [30,31].

From the evolutionary point of view, the *TP*53 gene is specific for the Holozoa branch, where its ancestral p63/p73-like genes emerged approximately one billion years ago [32,33]. The p53/p63/p73 protein family plays key roles in several major molecular and biological processes, including tumor suppression, fertility, mammalian embryonic development, and aging [20]. Unlike *TP*53, *TP*73 and *TP*63 genes are rarely mutated in cancers. Yet, the tumor suppressor function of p73 (tp73) is often attenuated in human cancers. The mechanism of suppression is via the hypermethylation of CpG islands at promoter 1, the binding to the overexpressed dominant-negative p73 isoform, dNp73 [34], or to E3 ubiquitin-protein ligase Mdm2 or p53-binding protein Mdm4. The pharmacological inhibition of protein–protein interactions is currently being explored for improved cancer therapy [35]. Notably, it was demonstrated that all p53 family members take part in regulating aging through the activation of senescence and regulating DNA repair [18,25].

p53 is a major factor that regulates cellular senescence, and the mechanism is via the activation of cyclin-dependent kinase inhibitor 1 *CDKN*1A (p21) and promyelocytic leukemia protein, PML. The study by Tyner et al. showed that heterozygous mice having one *TP*53 allele with the deletion of the first six exons (*Tp*53^+/m^, Δ exon 1–6) aged prematurely. These mutant mice exhibited enhanced resistance to spontaneous tumors, yet displayed accelerated aging compared to *Tp*53^+/+^ mice [36]. A study by the same group showed that truncated p53 protein stabilized wild-type p53 in non-stressed cells promoted its nuclear accumulation, and induced the hyperstability of wild-type p53 upon irradiation [37]. Based on this observation, the conclusion was that the constitutive expression of p53 accelerates aging. This was not confirmed in a follow-up study [38], as the pro-aging phenotype was not seen in the p53 “super-mice” expressing additional copies of the *TP*53 gene. Thus, over-activated p53 *per se* might not be a critical driver of accelerated aging. Yet, the role of p53′s hyperactivity in aging appears to be conflicting. Fibroblasts derived from hereditary segmental progeroid syndrome patients with the homozygous antiterminating mutation c.1492T > C in the *MDM*2 gene showed p53 hyperstability and accelerated aging [39]. This study postulated that the hyperstability of p53 due to an aberrant MDM2–p53 axis and the exposure to chronic stress induces the aging phenotype through the induction of chronic senescence. MDM2, the best-described negative regulator of p53, binds to the N terminal domain of p53 via its N-terminus. The knockout *Mdm*2^−/−^ mice show embryonic lethality in a wtp53 background, which indicates that p53 regulation by MDM2 is critical for development. Yet, the conditional deletion of *Mdm*2 in the epidermis induced p53-mediated senescence and accelerated aging [40]. Thus, a deregulated MDM2–p53 axis might play a role in the aging phenotype.

Gradual DNA damage and mitochondrial decline are hallmarks of physiological aging. DNA damage that is activated by telomere attrition in an aging cell induces p53 and mitochondrial dysfunction through the repression of the PPARγ co-activator 1α (PGC1α). This induces senescence [41]. Furthermore, a study on the hereditary segmental progeroid syndrome clearly highlighted the role of Mdm2 inactivation and p53 hyperactivity in the aging phenotype [39]. Despite the emerging evidence, the exact molecular mechanisms underlying the p53-mediated aging phenotype need to be elucidated. For example, it was demonstrated that replicative senescence is facilitated by p53, mainly through the activation of *CDKN*1A. Yet, several other factors contribute to aging, such as the activation of E2F and mTOR, as described elsewhere [18]. In principle, it can be concluded that p53 prevents cancer and protects from aging under physiological conditions; however, chronic-stress-amplified p53 has a detrimental effect on healthy aging despite retaining its tumor suppression function. Hence, p53 can either be a pro-aging or a pro-longevity factor, depending on the physiological context [42].

In addition to full-length p53, p53 isoforms may also play an important role in the modulation of longevity. The expression of certain short and long forms of p53 protein might contribute to a balance between tumor suppression and tissue regeneration [43]. For example, the p53β isoform, which is generated through the alternative splicing of intron 9, is upregulated in normal human senescent fibroblasts and interacts with full-length p53 to induce *CDKN*1A [44].

Considering the critical role of p53 in maintaining tissue homeostasis, high frequency of gain of function (GOF) mutations in cancer, and the limited and conflicting information on p53 role in organismal aging in Animalia, in the present work, we employed currently available datasets and tools to analyze p53 protein sequences in species possessing an extended lifespan. Our thorough analysis depicted a surprising correlation between the changes in the p53 protein sequence and the organismal lifespan, both in short- and long-lived species. Many of the identified changes occurred in the DNA-binding domain and might have a detrimental effect on p53 DNA-binding activity. Overall, we found that, when compared to the majority of closely related organisms within their phylogenetic groups, animals with unusually long lifespans share atypical p53 protein sequence features when compared to human p53 in the position corresponding to the human 180–192 p53 region, which points to the important contribution of loop 2 in the p53 core domain regarding life expectancy.

## 2. Results

The growing evidence implies that p53 activity might play a pivotal role in aging in humans and little is known about the molecular signatures of extended lifespan in animals including humans; thus, we inspected all currently available sequence data of long-lived animals to explore a link between longevity (maximal lifespan) and p53 protein sequences. For this, we used the longevity data from the AnAge Database [45]. We merged all available p53 protein sequences from the RefSeq database with the AnAge Database (for more detail, refer to the Materials and Methods section). The p53 sequence from 118 species and their lifespan data were cataloged and sorted according to their phylogenetic group (Appendix A).

The longest living animal in our dataset was the bowhead whale (*Balaena mysticetus*) from Artiodactyla (subgroup Cetacea), with a maximal lifespan of 211 ± 35 years [46]. Bowhead whales had a significantly longer lifespan (about four times longer) compared with other whales. A thorough comparison of p53 protein sequences showed that, in contrast to other Cetacea, *Balaena mysticetus* had a unique leucine substitution in the proline-rich region, corresponding to amino acid residue 77 in human p53 (Figure 1). Even though the change in the amino acids was predicted to be neutral according to the Protein Variation Effect Analyzer (PROVEAN) score of −0.993, the substitution still might change the activity of p53. Yet, this could only be addressed by extended functional studies. All other accessible p53 sequences of whales had an identical amino acid residue in this position to human p53.

Most amphibian species live for less than 30 years [45]; however, the olm (*Proteus anguinus*, Batrachia, Amphibians), which is the only exclusively cave-dwelling chordate, has a maximal documented lifespan of 102 years. A comparison of the p53 protein sequences in amphibians showed a previously unrecognized insertion in *Proteus anguinus*. The p53 protein from this species had additional serine and arginine residues in the core domain (corresponding to an insertion after amino acid L188 in human p53), which had a deleterious effect on p53 functionality according to the PROVEAN tool (Figure 2, Table 1).

The kakapo (*Strigops habroptila*, Aves) is a long-lived, large, flightless, nocturnal, ground-dwelling parrot that is endemic to New Zealand with a lifespan of around 95 years (Figure 3A, blue bar). A comparison of its p53 protein sequence with other related species showed a change at positions 128 and 131, corresponding to the following changes in human p53: P128V and N131H (Figure 3B). Interestingly, N131H mutations in human p53 are found in pancreatic and colon cancers [48,49]. This mutation most probably changes the structure of the p53 core domain and decreases the ability of p53 to bind to a canonical DNA sequence. Relevantly, according to the Phenotypic Annotation of *TP*53 Mutations (PHANTM) classifier, the N131H mutation decreases p53 transcriptional activity by 47.19% [50]. In addition, according to the PROVEAN tool, substitutions at position 128 were deleterious with a score of −4.45 (Table 1). These findings support the hypothesis that the change in p53 in the kakapo is linked to the loss of function. We speculate that the lack of exposure to sunlight, thus low incidence of UV-induced DNA damage, might render p53 inactive in this species.

Next, our analysis identified alterations in p53 in species having a long lifespan in the Chiroptera order. The Brandt’s bat (*Myotis brandtii*) is an extremely long-lived bat with a documented lifespan of 41 years [51]. Together with its close relative *Myotis lucifugus*, they had significantly longer lifespans than other bats (Figure 4A, blue bars). These two species share a unique arrangement in the p53 DNA-binding region, with the insertion of seven amino acid residues in the central DNA-binding region (following amino acid 295 in the human p53 canonical sequence) (Figure 4B). To assess how this rearrangement in the DNA-binding region changes the interaction of p53 with DNA, we next modeled the p53 tetramer using the SWISS-MODEL workflow. The insertion in the DNA-binding domain of bats with a long lifespan occurred in the DNA interaction cavity, suggesting the decreased affinity of p53 for binding to DNA (Appendix A). *Myotis brandtii* and *Myotis lucifugus* are very small bats (max 8 g body weight) and provide a significant exception from Max Kleiber’s law (mouse-to-elephant curve) since their lifespan is extremely long in relation to their small body size [52].

The abovementioned analysis of long-lived organisms in various animal groups led us to conclude that the amino acid sequence of p53 was associated with organismal lifespan. Therefore, we continued the analysis by further correlating the p53 amino acid sequences with the lifespan across the animal kingdom. Due to the low similarity between the p53 N-terminal and C-terminal domains across species and the significant role of mutations in the p53 DNA-binding domain in cancer, we focused on the most conserved core domain of p53 and constructed the p53-based tree (Figure 5, left panel) [32]. We then compared the contemporary phylogenetic tree with the tree based on the p53 protein sequence (Figure 5). Then, the dataset with p53 sequences and animal lifespans were divided into 12 groups based on their phylogenetic relationships. Interestingly, some p53 sequences were not closely associated with the phylogenetic tree, indicating several parallel evolutionary processes leading to modified p53 activity. Even closely related species in various groups had significantly different lifespans (Appendix A), and therefore, were suitable for correlation analyses according to the method introduced by Jensen and colleagues [53].

Figure 6 summarizes the lifespan data and the total number of analyzed animals for each group, with minimal and maximal values (shown in Appendix A). Only datasets with more than five members in the group were used in the correlation analyses.

The organisms with the longest lifespan in the Neopterygii dataset were the carp (*Cyprinus carpio* (47 years)), followed by the goldfish (*Carassius auratus* (41 years)). The Siamese fighting fish (*Betta splendens)* had the shortest lifespan in the group (2 years). The correlation analyses show that fifteen amino acid residues in the p53 core domain were significantly associated with a prolonged lifespan (Figure 7A). We found that the most common variation in the long-lived Neopterygii was the presence of a serine (S) at positions corresponding to 98, 128, and 211 of human p53, and the presence of valine (V) at positions 128, 150, 217, and 232. On the other hand, in the short-lived organisms in Neoropterygii, we identified threonine (T) at positions 98, 100, 141, 217, and 260; glutamic acid (E) at positions 110, 128, 150, and 291; and serine at positions 141, 203, and 235. We reasoned that the abundance of glutamic acid could result in the decreased affinity of p53 to DNA due to the local change in the ionic charge at the site of the amino acid p53 variant. Indeed, the PROVEAN tool predicted a deleterious effect on p53 function for glutamic acid at position 128. In addition, according to the PHANTM classifier, C141S substitution led to a decrease in p53 transcription activity by 41.08% as compared to wtp53.

The lifespans of species in Sauria were significantly variable. The organisms with the longest lifespan in this group were the three-toed box turtle (*Terrapene carolina triunguis* (138 years)) and the kakapo (*Strigops habroptila* (95 years)). The green anole (*Anolis carolinensis*) had the shortest lifespan in the group (7.2 years). The correlation analyses showed that, similar to Neopterygii, a specific fifteen-amino-acid-residue fragment in the p53 core domain was associated with a prolonged lifespan (Figure 7A). The most common p53 variation in long-lived Sauria was similar to Neopterygii and it was the higher abundance of serine (corresponding to positions 94, 95, 149, and 227 in human p53) and the presence of valine (at positions 97 and 232, identical to Neopterygii). When compared to human p53, in short-lived organisms, we identified threonine at positions 94, 149, 159, and 227, and glutamic acid at positions 114, 192, and 228. In addition, deletions in the p53 sequence were found at positions 94–97 and 114 (Figure 7A). Similar to Neopterygii, the most common p53 variation in short-lived Sauria was the presence of threonines and glutamic acid residues. However, more studies are needed to elucidate the functionality of these p53 sequences.

The organisms with the longest lifespans in the Primates group were humans (*Homo sapiens* (122 years)) and the western gorilla (*Gorilla gorilla* (60 years)). Tarsier (*Carlito syrichta)* had the shortest lifespan in the group (16 years). The correlation analyses showed that the specific amino acid triad—Q^104^, S^106^, L^289^—was significantly associated with a prolonged lifespan (Figure 7A). Besides serine at position 106, two others—glutamine (Q) at position 104 and leucine (L) at position 289—are both hydrophobic and might impact the structure of the DNA-binding domain.

In contrast, proline (P) or histidine (H) at position 104, asparagine (N) at position 106, and phenylalanine (F), serine (S), or tyrosine (Y) at position 289 were associated with short-living primates. While studying human longevity, one needs to consider that the prolonged lifespan of *Homo sapiens* is associated with cultural and socio-economical advantages. Therefore, we performed additional analyses after excluding *Homo sapiens* from the dataset. The same variations were observed in the correlation analyses. Taken together, our results demonstrate that the amino acid variations shown in Figure 7A were conserved in the following closely related species: *Homo sapiens*, *Pan troglodytes*, and *Gorilla gorilla*.

The dataset of Glires contained seventeen species with lifespans ranging from 3.8 to 31 years. The organisms with the longest lifespan in this group were *Heterocephalus glaber* (31 years) and *Castor canadensis* (23). The shortest lifespan in the group was *Rattus norvegicus* (3.8 years). The correlation analyses showed that ten amino acid residues were significantly associated with a prolonged lifespan (Figure 7A). Two threonine residue variations (positions 123 and 210) were present in long-lived Glires. Other amino acid changes occurred only once. Interestingly, in short-lived Glires, there was also a significant presence of threonine at two other locations (positions 148 and 150). Similar variations were also observed in the methionine residues (at positions 123 and 201), tyrosine (positions 202 and 229), and proline (positions 185 and 201).

The organisms with the longest lifespan in the dataset of Chiroptera were the brandt bat (*Myotis brandtii* (41 years)) and the little brown bat (*Myotis lucifugus* (29 years)). The pale spear-nosed bat (*Phyllostomus discolour)* had the shortest lifespan (9 years). The correlation analyses showed that nine-amino-acid-long motif was associated with a significantly prolonged lifespan (Figure 7A).

The organisms with the longest lifespans in the Carnivora group were the polar bear (*Ursus maritimus*) and the panda (*Ailuropoda melanoleuca*). The shortest lifespan in the group was that of the ferret (*Mustela putorius furo*). The correlation analyses showed that two amino acids in the p53 core domain, position 148 and 232, were significantly associated with lifespan (Figure 7A). While the presence of asparagine at position 148 and valine at position 232 were associated with a long lifespan, the presence of a serine at position 148 and isoleucine at position 232 was associated with a short lifespan.

The organism with the longest lifespan in the Artiodactyla group was the bowhead whale (*Balaena mysticetus* (211 years)) followed by the orca (*Orcinus orca* (90 years)). Correlation analyses showed that twelve amino acid residues in the p53 core domain were significantly associated with prolonged lifespan (Figure 7A). Similar to Neopterygii and Sauria, the most common variation present in the long-lived organisms were associated with high abundance of serine (corresponding to positions of 106, 148, and 166 in human p53). The variations of serine at positions 129, 182, and 222 were the most common variations for the short-lived Artiodactyla, together with variations in the glutamic acid residues at positions 226 and 228.

Next, we investigated all 118 RefSeq p53 sequences to evaluate the associations between amino acid variations and maximal lifespan (Figure 7A). When applying the Bonferroni correction, only two significantly associated residues were revealed, corresponding to human serine (S) 185 and asparagine (N) 210. Organisms that have serine at position 185 live statistically longer than organisms with another amino acid in this position. Interestingly, p53 S185 variants are rare in humans and only a few variants were found to be cancer-specific, suggesting that S185 might be a conserved amino acid that is critical for organismal longevity [54]. On other hand, organisms that contained glutamine (Q) instead of asparagine (N) at position 210 had a significantly shorter maximal lifespan. Without a Bonferroni correction, from the 200 analyzed positions of the aligned p53 core domains (related to human p53 94–293 aa), 64 positions were significantly associated with lifespan (Figure 7B). Positive correlations with longevity are shown using orange and red colors, green and blue show negative correlations. However, more detailed studies are needed to fully apprehend the functionality of the changed p53, both in the short-lived and in the long-lived organisms.

The changes at the molecular level are often a result of the adaptation of species to environmental forces. To evaluate whether the amino acid residues in the p53 core domains (aa 94—293 of the human p53 canonical sequence) share some relevant features in relation to the convergent evolution, we constructed a sequential circular representation of the multiple sequence alignments and the mutual information it contains (Figure 8A). The figure shows that the amino acid residues that were significantly associated with longevity (extracted from the heatmap (Figure 7B), highlighted in light green) very often coevolved together (represented by connected lines). This observation may provide evidence for the convergent evolution of p53 proteins in organisms with extreme longevity. According to Passow and colleagues, taxa with evidence of positive selection in the *TP*53 gene are those with the lowest incidences of cancer reported in amniotes (elephants, snakes, lizards, crocodiles, and turtles) [55].

The longest region in p53 that was associated with longevity spanned amino acids in loop 2 (L2) of human p53 (Figure 8A, green, dashed circle, residues 180–192). L2 is the minor groove binding region of human p53 and the stability of this region is maintained by Zn^2+^. The loss of Zn^2+^ triggers the aggregation of L2 and the loss of DNA-binding specificity. The 3D structures of DNA binding domain (DBD) in selected long-lived species revealed intrinsic variations in L2 structure when compared to human p53 (Figure 8B). It is possible that intrinsic changes in L2 due to amino acid changes in long-lived species amend p53 DNA-binding specificity and/or the binding of co-factors via an allosteric mechanism and alter the p53-driven senescence program.

Table 1 lists the species characterized by the extreme longevity together with the associated p53 variations identified in our study. Apart from the unique substitutions (*Strigops habroptila*, *Balaena mysticetus*) and insertions (*Myotis brandtii*, *Myotis lucifugus*, *Proteus anguinus*), a complete lack of p53 mRNA expression was found in *Turritopsis* sp.

To gain a better insight into the putative changes in the p53 regulatory pathways in the long-lived species in which the p53 protein remains unaltered, we analyzed the sequence of p53 regulators. SIRT1 deacetylates p53 in an NAD^+^-dependent manner and inhibits p53 transcription activity [56]. We found that SIRT1 had an atypical protein sequence in *Cebus imitator*, a model organism for studying extreme longevity in primates, where the amino acid sequence is different from all other primate SIRT1s. 3D modeling revealed that the predicted structure of *Cebus imitator*’s SIRT1 was significantly different from the structures of SIRT1 from *Homo sapiens* and *Sapajus apella* (a close relative of *Cebus imitator*) (Figure 9).

We hypothesized that SIRT1 from *Cebus imitator* gained new functions, which might result in the decreased activity of p53 when compared to other primates and slow down the aging processes, most likely via the transient inhibition of p53. Yet, it remains to be elucidated which factors might be affecting the altered target recognition by SIRT1.

In addition to SIRT1, we investigated other key factors in the p53 pathway. Surprisingly, we found that *Myotis brandtii* (long-living bat described above), in addition to p53, had two atypical protein sequences, one in UFM1 (Ubiquitin-fold modifier 1, XP_005862786.1) and the other in the p73 (tumor protein 73, XP_014401672).

Recently, it was reported that UFM1 covalently modifies p53; this phenomenon is called UFMylation [57]. UFMylated p53 is stabilized at the protein level, as this covalent modification antagonizes p53 ubiquitination and proteasomal degradation.

In UFM1, we found an approximately 20-amino-acid-long extension of the C-terminal end in *Myotis brandtii* (and also in two other myotis bats—*Myotis lucifugus* and *Myotis myotis*) (see Figure 10). In contrast, in other bats that live much shorter (e.g., the closest myotis bats relative is *Pipistrellus kuhlii*, with a maximal lifespan of only 8 years) and in the rest of mammals, including humans, no such extension occurs. We hypothesize that the extended UFM1 protein might contribute to the extreme longevity in myotis bats through the loss of function and consequent changes in p53 protein degradation patterns. Yet, more experimental evidence is needed to draw a clear conclusion.

Lastly, we found unique changes in the p73 protein sequence in the *Myotis brandtii* bat (XP_014401672). There were multiple large deletions (>10 amino acid residues) in critical p73 regions, which were found exclusively in this extremely long-lived bat. p73 is the transcription factor that undergoes similar cellular regulation as p53 protein and its role in aging is attributed to the induction of senescence through the upregulation of *CDKN*1A [58].

Taken together, our analyses revealed the unexpected correlation between p53 sequence variations and longevity in the animal kingdom. The changes may affect p53 functionality and, thus, influence the activation of replicative senescence, a hallmark of molecular aging. In long-lived species, with no changes in p53, the upstream regulatory proteins, including SIRT1 and UFM1, displayed amino acid changes that may affect their functionality and in consequence alter p53 activity. Yet, further studies are needed to fully comprehend the role of amino acid changes in p53 and its role in the long-lived species described in our work.

## 3. Discussion

The p53 protein is a well-known tumor suppressor and *TP*53 is the most often mutated gene in human cancers. On the cellular level, in humans, increased p53 activity protects from cellular stress and enables genome stability, whereas altered mutant p53 protein functionality is essential for cells’ immortalization and neoplastic transformation [59]. However, the role of variations in the p53 amino acid sequence at the organism level in other animals has not been studied systematically. Here, we addressed the role of p53 in longevity in the animal kingdom by presenting an in-depth correlation analysis manifesting the dependencies between p53 variations and organismal lifespan. To date, p53 expression has been detected in all sequenced animals from unicellular Holozoans to vertebrates [32]. The seminal work by Kubota provided important evidence demonstrating that immortality is not just a hypothetical phenomenon. He demonstrated that the Cnidarian species *Turritopsis* jellyfish is immortal and can repeatedly rejuvenate, reverse its life cycle, and thus, was the first and only known “immortal” animal on Earth [60]. Here, we inspected recently published data from the whole-transcriptome data of “immortal” *Turritopsis* sp. [61] and surprisingly found no expression of any of the p53 protein family members in the pooled data from all individuals at all developmental stages (polyp, dumpling with a short stolon, dumpling, and medusa). This points to the possibility that the absence of p53 in *Turritopsis* might be directly related to its unique ability of life cycle reversal and “immortality.”

Telomere shortening in humans induces replicative senescence, which is a process that is regulated by p53. In the absence of p53, the replicative lifespan of human cells is extended and the concurrent loss of retinoblastoma protein (RB) extends the replicative lifespan to a greater extend (reviewed in [26]).

Intriguingly, our results obtained using Protein Variation Effect Analyzer [62] show that the variability in lifespan among closely related species correlated with specific p53 amino acids’ variations. Long-lived organisms were characterized by specific substitutions in the p53 amino acid sequence. It is likely that the amino acid changes imposed on p53 in long-lived species enable p53 to interact with different multiple protein partners to induce gene expression programs that vary from those induced in species with a relatively normal lifespan.

We identified the 180–192 region, corresponding to the loop 2 (L2) region of human p53, as the longest region that is associated with longevity. The 3D model revealed variations in L2 structure in long-lived species when compared to human p53. Loop 2 is responsible for binding to the minor groove and its structure depends on the presence of Zn^2+^. We speculate that in long-lived species, L2 affects the p53 binding to DNA and/or other transcription factors and, consequently, affects the replicative senescence program. On other hand, in humans, the L1 region is responsible for p53 binding to the major groove and was reported to undergo the most dynamic changes among the DNA contacting loops (L1–L3) when located on a non-target or target DNA sequence [63]. Our findings indicate that the L2 region, but not L1, might play a role in modulating the senescence (or other pro-aging program) in long-lived species. Yet, detailed functional studies are needed to fully comprehend the role of p53 alterations in longevity.

Based on what is known about the processes underlying aging, we anticipate that the altered gene expression programs would enable the following changes (Figure 11): (1) more efficient tissue repair through autophagy, (2) loss of replicative senescence, (3) enhanced clearance of senescent cells by the immune system, (4) enhanced regulation of intracellular ROS levels, (5) improved resistance of mitochondria to ROS-induced damage, or (6) loss of immune senescence that occurs in humans during healthy aging. All of the above processes were previously described as significantly contributing to longevity in humans (reviewed in [18]). Intriguingly, a recent GWAS study on 1 million parent lifespans identified only several variants influencing lifespan at genome-wide significance, including *CDKN2B-AS*1 and *IGF2R*. The *TP*53 gene was not among the singled-out variants, which, in accordance with our observations, indicates that no changes in human p53 might be attributing to longevity in humans [64]. Our analysis demonstrates that the long-lived organisms might have different mechanisms of protection against cancer that are not directly linked to p53 activity. We speculate that their lifespan is not limited by somatic cells’ senescence caused by the chronic stress-induced hyperactive p53 protein, which is the case for other species with shorter lifespans (Figure 11).

The maximal lifespan according to the AnAge database is attributed to the Greenland shark, with an estimated maximal life span of 300–500 years. Unfortunately, no transcriptomic nor genomic data for the Greenland shark (*Somniosus microcephalus*) are available. Compared to other sharks (with a life expectancy of up to seventy years), its lifespan is exceptional. It will thus be very interesting to know the sequence of their p53 protein. A recent study suggests that certain animals may have evolved to have longer lifespans compared to other species belonging to the same taxa [65]. In addition, the authors found that the outliers among taxa (in terms of maximum age) always had longer lifespans, not shorter. This would support our hypothesis that extreme longevity is a result of adaptive mutational changes in the particular critical gene(s), allowing the organisms to escape the senescence machinery.

Experimental data support our hypothesis that specific p53 variations are associated with longevity. For example, it was found that the reduced expression of the *Caenorhabditis elegans* p53 ortholog, namely, cep-1, results in increased longevity [66]. It was also demonstrated that the neuronal expression of p53 dominant-negative proteins in adult *Drosophila melanogaster* inhibits the function of full-length p53 and extends their lifespan [67]. The same principle is most probably present in humans, where, for example, p53 variants that predispose to cancer are present in healthy centenarians [68] and a meta-analysis showed that the codon 72 polymorphic variant of p53 with proline (compared to arginine) was associated with increased cancer risk and with the increased survival [69]. In a recent study by Zhao et al., polymorphism at position 72 (P72 compared to R72) was reported to have a positive effect on lifespan and to delay the development of aging-related phenotypes in mice, supporting a role of the changed p53 activity in longevity [70]. Another example of a long-lived vertebrate is the elephant, which has 20 copies of the *Tp53* gene [71]. In this species, part of the DNA-binding region of p53 is deleted in all but one of the *TP*53 gene copies, which may result in the formation of dysfunctional p53 tetramers, thus presumably modulating p53 transcriptional activity in response to stress [71]. In contrast, a study by Tejada-Martinez et al. [72] in cetaceans did not single out *TP*53 as a gene associated with extreme longevity. Yet, the authors provided evidence that natural selection in tumor suppressor genes (including *TP*53) could act on species with an extended lifespan. In the naked mole rat (*Heterocephalus glaber*), which is the longest-living small rodent and weighs only around 35 g, a unique hyperstabilization and nuclear accumulation of the p53 protein were recently reported [73]. The naked mole rats’ natural habitat is in the hypoxic environment underground in constant darkness. Despite their extremely long lifespan of up to 30 years, naked mole rats show very little biological decline, neurodegeneration, and senescence [74]. The hyperstability of the naked mole rat’s p53 when compared to murine p53, which is independent of genotoxic stress, might be a consequence of the change in the amino acid sequence in the p53 protein. Yet, an in-depth analysis must be performed to decipher the mechanisms leading to the unprecedented stability of p53 in the naked mole rat and to understand the role of hyperstable p53 in the longevity in this species.

Intriguingly, the animals with extreme longevity that we have identified are mostly nocturnal or live in absolute darkness. These include *Strigops habroptila*, *Myotis brandtii*, *Myotis lucifugus*, *Proteus anguinus*, *Balaena mysticetus*, and *Heterocephalus glaber*. It is thus likely that low or no exposure to the UV light promotes the evolutionary changes in the p53 protein structure that alter the p53′s pro-senescence activity. In addition, we speculate that in those species, the endogenous levels of reactive oxygen species might be lower when compared to animals from other, less extreme habitats. This might be a consequence of the changes in the metabolism rates that might affect the overall rate of oxidative phosphorylation and, consequently, slow down the generation of free radical species through the electron transport chain. This hypothesis agrees with the recent study showing that rapamycin, which is a widely studied inhibitor of mTOR, prevents UV-induced skin aging through the inhibition of p53, reversal of UVA-induced cellular senescence, and induction of autophagy [75].

Despite the high complexity of the p53 proteins family, modern methods of comparative genomics provide useful tools for exploring protein variations in closely related species and correlating the extracted molecular information with lifespan [76]. According to Sahin and DePinho, the hyperactivity of p53 in the presence of accumulated DNA damage and ROS is one of the main causes of aging [41]. This observation is in congruence with our hypothesis that organisms with atypical p53 sequences, likely attenuating the wtp53 activity, are extremely long-lived. Of note, even though several p53 amino acid changes were found in various animal groups, some variations developed in convergent evolutions in different groups of species. For example, the presence of threonine and glutamic acid was observed in short-lived organisms of different groups, and the richness of serine residues was typical for long-lived organisms in several groups. Next, a serine residue at position 185 was significantly associated with a prolonged lifespan across all analyzed species. Yet, further mechanistic studies are needed to pin down how the identified p53 changes affect its functionality, how the amino acid changes contribute to longevity, and how this knowledge can be translated for prolonging healthy aging in humans.

## 4. Materials and Methods

### 4.1. Searches of Maximal Lifespan

To access data on longevity and maximal lifespan, we used the AnAge Database (https://genomics.senescence.info/species/, accessed on 4 May 2020); AnAge currently contains data on the longevity of more than four thousand animals [45]. We downloaded the whole dataset and selected species that were presented in the NCBI RefSeq database.

### 4.2. Protein Similarity Searches

For the protein similarity searches, we downloaded all available p53 sequences from the RefSeq database (https://www.ncbi.nlm.nih.gov/refseq/, accessed on 10 October 2020) and merged them with the AnAge Database. We received the p53 sequence information of 118 species with information about their lifespan and sorted them according to their phylogenetic group (Appendix A). For animals with extreme longevity, where the p53 homologs were not present in the NCBI database, local BLAST searches (tblastn) applied to de novo assembled transcriptomes were used together with the default “BLAST+ make database” command and searching parameters within the UGENE standalone program [77].

### 4.3. Transcriptome Assemblies

Transcriptomic data for the bowhead whale was obtained from http://www.bowhead-whale.org/, accessed on 3 July 2020 [46]. When there were only raw seq reads from the RNA-seq experiments available (deposited in the NCBI SRA), we performed the de novo assembly first using the Trinity tool [78] from the Galaxy webserver (https://usegalaxy.eu/, accessed on 7 September 2020) [79] with default settings. This was done for *Proteus anguinus* (SRX2382497) and *Sphenodon punctatus* (SRX4014663); the resulting assemblies are enclosed in Appendix A.

### 4.4. p53 Protein Tree and Real Phylogenetic Tree Construction

The protein tree was built using the Phylogeny.fr platform (http://www.phylogeny.fr/alacarte.cgi, accessed on 15 October 2020) [80,81] and comprised the following steps. First, the sequences were aligned with MUSCLE (v3.8.31) [47], which was configured for the highest accuracy (MUSCLE with default settings). After the alignment, ambiguous regions (i.e., containing gaps and/or poorly aligned) were removed with Gblocks (v0.91b) [82] using the following parameters: minimum length of a block after gap cleaning: 10; no gap positions were allowed in the final alignment; all segments with contiguous non-conserved positions longer than 8 were rejected; minimum number of sequences for a flank position: 85%. The phylogenetic tree was constructed using the maximum likelihood method implemented in the PhyML program (v3.1/3.0 aLRT) [83,84]. The JTT substitution model was selected by assuming an estimated proportion of invariant sites (of 0.204) and 4 gamma-distributed rate categories to account for the rate heterogeneity across sites. The gamma shape parameter was estimated directly from the data (gamma = 0.657). Reliability for the internal branch was assessed using the bootstrapping method (100 bootstrap replicates). Graphical representation and editing of the phylogenetic tree were performed with TreeDyn (v198.3) [85]. The real phylogenetic tree was reconstructed using PhyloT (https://phylot.biobyte.de/, accessed on 4 September 2020) and visualized in iTOL (https://itol.embl.de/) [86].

### 4.5. Prediction and Statistical Evaluation Using PROVEAN

The effect of the p53 variations in long-lived organisms was predicted and statistically evaluated using the Protein Variation Effect Analyzer web-based tool (PROVEAN; http://provean.jcvi.org/index.php, accessed on 20 May 2021) [62,87]. PROVEAN is a software tool that predicts whether an amino acid substitution or in/del has an impact on the biological function of a protein [62]. All inspected p53 variations in selected animals were statistically evaluated and numbered according to the human canonical p53 sequence (NP_000537.3).

### 4.6. Modeling of 3D Protein Structures

We used the SWISS-MODEL template-based approach (https://www.swissmodel.expasy.org/interactive, accessed on 3 March 2021) [88] to predict the 3D structures using individual FASTA sequences and reference PDB:4mzr as the crystal structure of the p53 tetramer from *Homo sapiens* with bound DNA [89]. The resulting PDB files are enclosed in Appendix A. The predicted p53 structures were visualized in UCSF Chimera 1.12 [90]. Effects of the novel mutation on SIRT tertiary structure were predicted using RaptorX [91].

### 4.7. Correlation of the Maximal Lifespan and Alterations within the p53 Core Domain in Vertebrates

Residue level genotype/phenotype correlations in p53 multiple sequence alignment were performed using SigniSite 2.1 (http://www.cbs.dtu.dk/services/SigniSite/, accessed on 22 October 2020) [53] with a significance threshold *p*-value of ≤0.05. A Bonferroni single-step correction for multiple testing was applied for the global correlation of all sequences, no correction was applied for smaller groups of taxonomically related animals. The manually curated set of 118 high-quality p53 protein sequences obtained from the NCBI (https://www.ncbi.nlm.nih.gov/, accessed on 20 July 2020) was used as the input file. These sequences were taken from the RefSeq database and the canonical isoform corresponding to the human full-length p53 isoform (NP_001119584.1) was manually filtered for each vertebrate species. The resulting set of these 118 p53 sequences was aligned within the UGENE workflow [77] and the MUSCLE algorithm [47] with default parameters. All sequences were then manually trimmed to preserve only the core domain, which corresponded to human 94–293 aa. Then, the numerical values of the maximal lifespan of each organism were added into the resulting FASTA file based on the information in the reference AnAge database (http://genomics.senescence.info/species/, accessed on 5 September 2020) [45].

### 4.8. Convergent Evolution

Multiple sequence alignments of the p53 core domains from 118 species were uploaded to the MISTIC webserver (http://mistic.leloir.org.ar/index.php, accessed on 18 November 2020), with PDB 2ocj (A) as the reference and using default parameters [92].

### 4.9. Gene Gain and Losses

*TP*53 gene gain or losses were inspected using Ensembl Comparative Genomics toolshed [93] via Ensembl web pages and *TP*53 gene query ENSG00000141510: https://www.ensembl.org/Homo_sapiens/Gene/SpciesTree?db=core;g=ENSG00000141510;r=17:7661779-7687550, accessed on 8 December 2020.

## 5. Conclusions

This study revealed a previously overlooked correlation between longevity and changes in p53 function due to the amino acid variations in the animal kingdom. Strikingly, several long-lived species, including *Myotis brandtii*, *Myotis lucifugus*, *Balaena mysticetus*, *Heterocephalus glaber*, *Strigops habroptila*, and *Proteus anguinus* displayed unique p53 protein sequence properties that were not shared with their close relatives that have a shorter lifespan. Altogether, our data show the convergent evolution of p53 sequences supporting a higher insensitivity to p53-mediated senescence under prolonged stress conditions in long-lived vertebrates. Our observations that specific variations of p53 protein are correlated with lifespan provide important grounds for the further exploration of p53 sequences in species displaying extreme longevity. Most importantly, our data implies a general mechanism at work in all vertebrates that leads to extended lifespan, which might be translated to studies on the extension of the health span in humans.

## Figures and Tables

**Figure 1 ijms-22-08512-f001:**
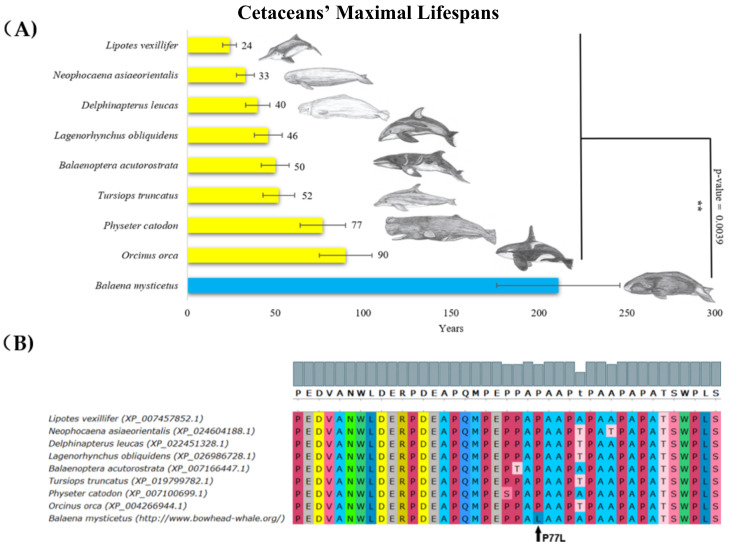
**Lifespan of species in the Cetacea order and the corresponding p53 sequence changes.** (**A**) Comparison of cetaceans’ maximal lifespans in years. The bowhead whale’s (*Baleana mysticetus*’s) maximal lifespan was more than twice the maximal lifespan of the rest of Cetacea (Wilcoxon one-sided signed-rank test was used, ** *p*-value < 0.01). (**B**) Multiple sequence protein alignments of p53 proline-rich region, performed in MUSCLE with default parameters [47], colors in “UGENE” style.

**Figure 2 ijms-22-08512-f002:**
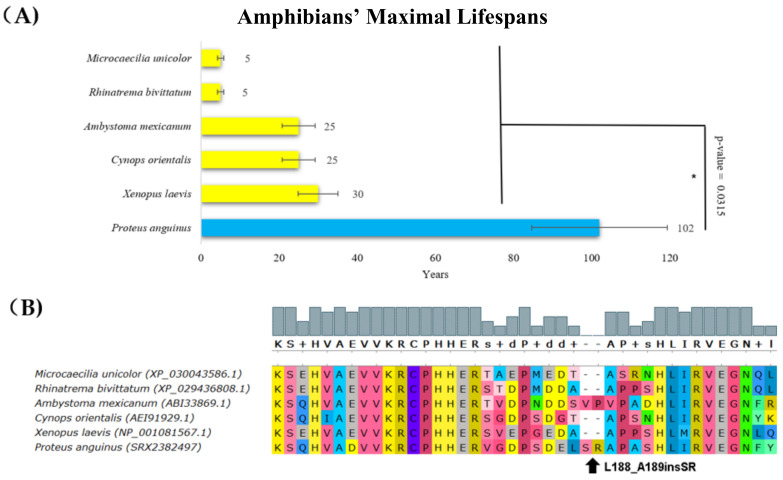
**Lifespan of species in amphibians and the corresponding p53 sequence changes.** (**A**) Comparison of amphibians’ maximal lifespans in years. The olm’s (*Proteus anguinus*’s) maximal lifespan was more than three times higher than the maximal lifespan of other amphibians (Wilcoxon one-sided signed-rank test, * *p*-value < 0.05). (**B**) Multiple protein alignments of the p53 dimerization region. The olm (*Proteus anguinus*) had an insertion that is two amino acid residues long following amino acid residue 188 (related to human p53 canonical sequence). The sequence of the p53 homolog from *Proteus anguinus* was determined using transcriptomic data from the SRA Archive (SRX2382497). The methods and color schemes are the same as in Figure 1B.

**Figure 3 ijms-22-08512-f003:**
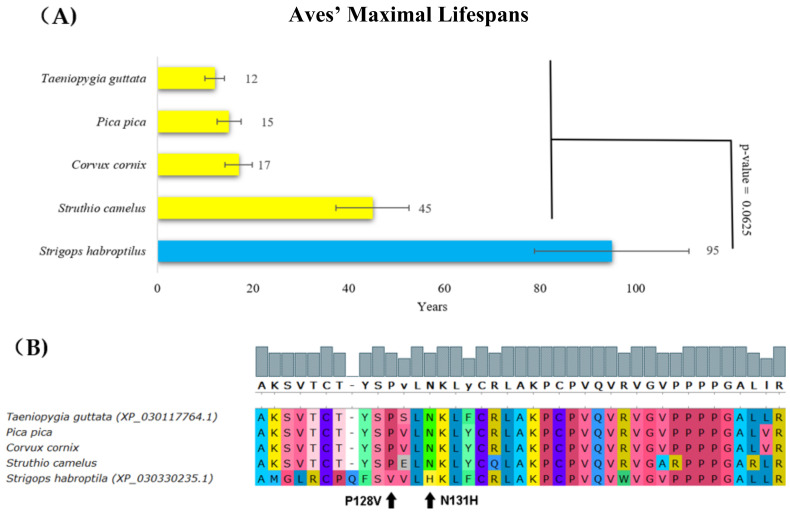
**Lifespans of species in the Aves order and the corresponding p53 sequence changes.** (**A**) Comparison of Aves’ maximal lifespans in years. The kakapo’s (*Strigops habroptila*’s) maximal lifespan was more than twice the maximal lifespan of other Aves (Wilcoxon one-sided signed-rank test). (**B**) Multiple protein alignments representing the partial p53 core domain of the accessible Aves sequences. Sequences of all avian p53 homologs were determined using transcriptomic data from the SRA Archive, except for *Strigops habroptila*, where the p53 sequence was known (XP_030330235.1). The methods and color schemes are the same as in Figure 1B.

**Figure 4 ijms-22-08512-f004:**
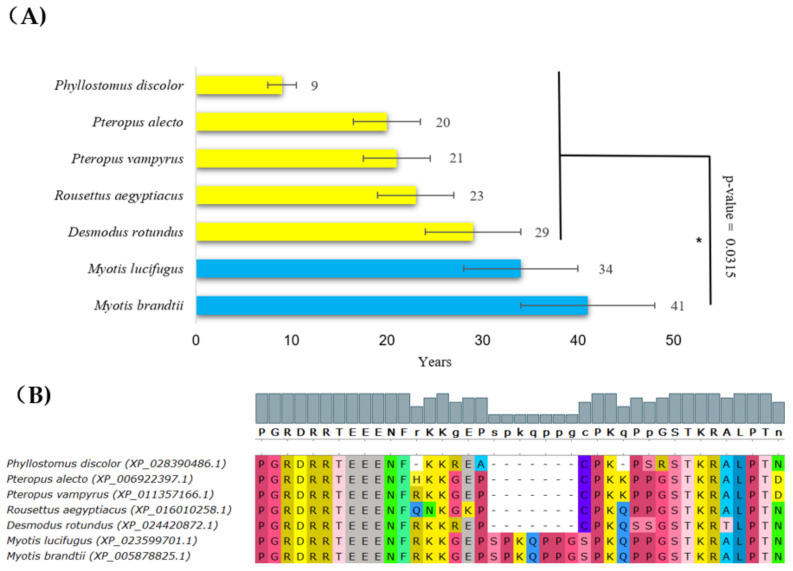
**Lifespans of species in the Chiroptera order and the corresponding p53 sequence changes.** (**A**) Comparison of Chiropteras’ maximal lifespans in years. The bats’, *Myotis brandtii*’s and *Myotis lucifugus*’s maximal lifespans were significantly longer compared with other sequenced bats (Wilcoxon one-sided signed-rank test, * *p*-value < 0.05). (**B**) Multiple protein alignments of the C-terminal part of the p53 core domain of accessible Chiroptera sequences. Methods and color schemes are the same as in Figure 1B.

**Figure 5 ijms-22-08512-f005:**
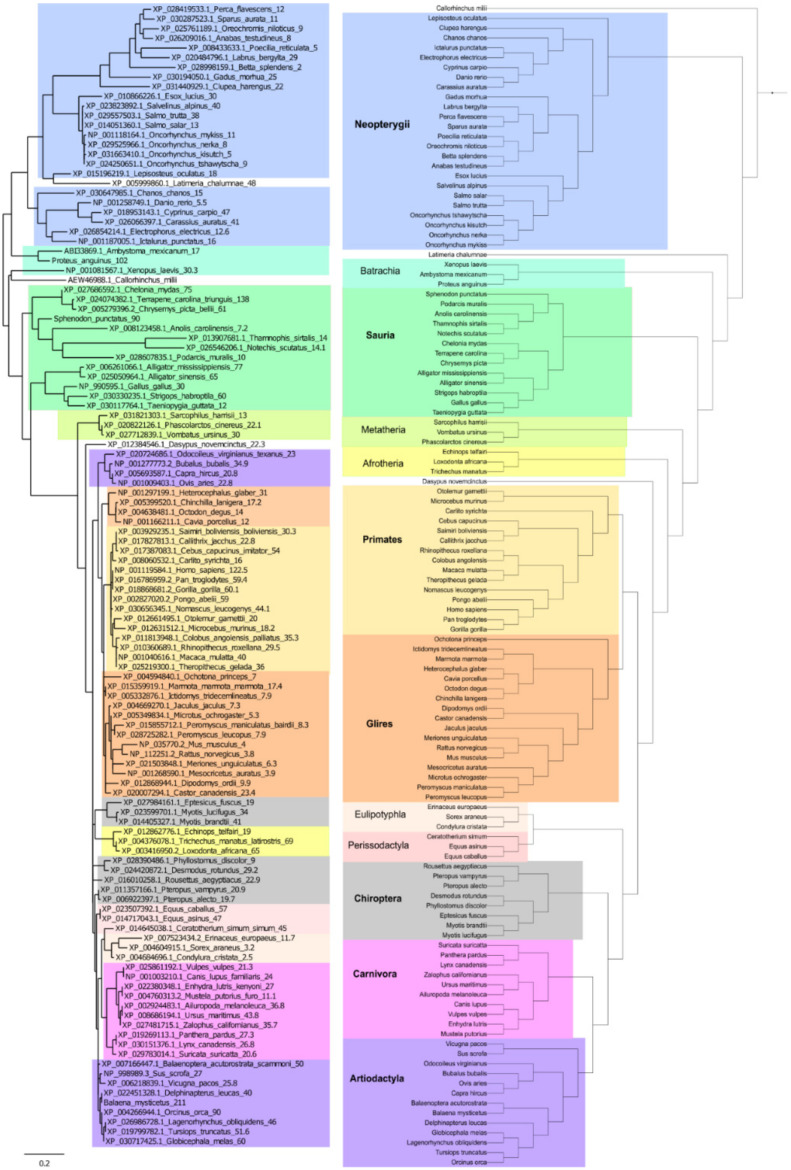
**The p53-based and contemporary phylogenetic trees.** Comparison of the p53 protein tree (**left**) and the real phylogenetic tree (**right**). The protein tree was built using the Phylogeny.fr platform. Organismal phylogeny was reconstructed using PhyloT and visualized in iTOL (see the Materials and Methods section for details). The same color backgrounds represent the same phylogenetic groups.

**Figure 6 ijms-22-08512-f006:**
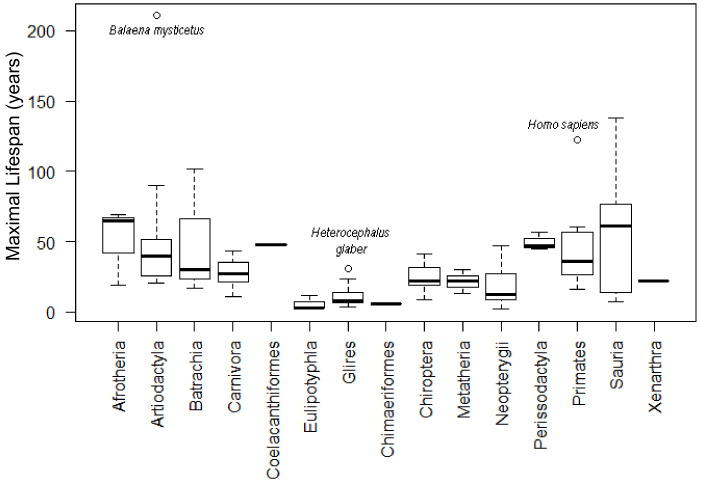
Representation of lifespans for all tested phylogenetic groups.

**Figure 7 ijms-22-08512-f007:**
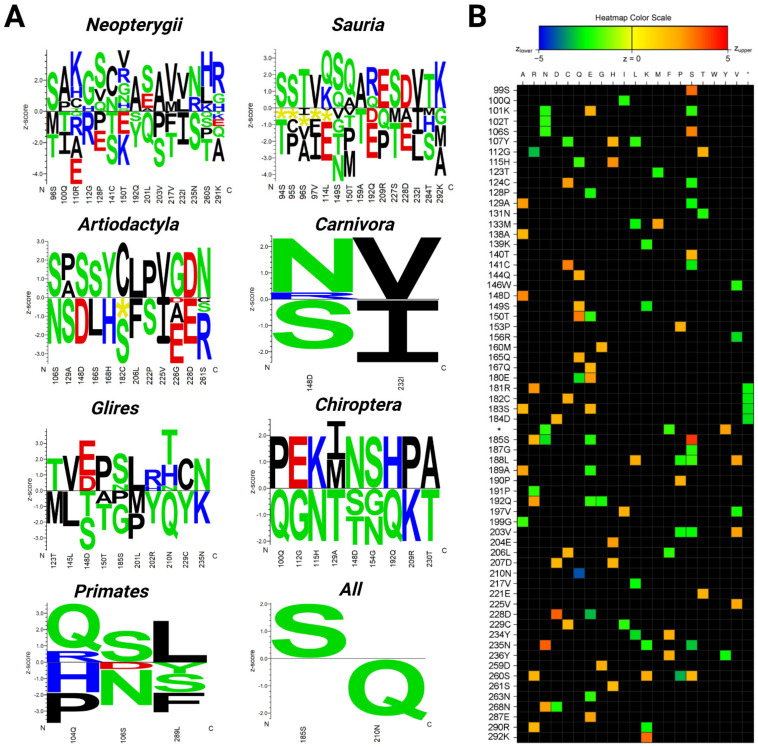
**Correlation of the most commonly altered p53 amino acid residues with the maximal lifespans of the analyzed species.** (**A**) Logos quantifying the strength of the p53 core domain residue association (related to the human aa 94–293 according to the p53 canonical sequence) with the maximal lifespan in years in the analyzed subgroups of animals. Amino acid residues on the positive y-axis were significantly associated with the prolonged lifespan phenotype and residues on the negative y-axis were significantly associated with the shorter lifespan phenotype (significance threshold *p*-value ≤ 0.05). The height of each letter representing the strength of the statistical association between the residue and the data set phenotype. The amino acids are colored according to their chemical properties as follows: acidic (DE): red, basic (HKR): blue, hydrophobic (ACFILMPVW): black, and neutral (GNQSTY): green. (**B**) Heatmap visualization of the strength of the residue association (without a Bonferroni correction). The color scale ranges from blue (z < −5) to red (z > 5). Each column corresponds to one of the 20 proteinogenic amino acids and each row to a position in the submitted multiple sequence alignment (Appendix A). * Indicates site of the amino acid insertion.

**Figure 8 ijms-22-08512-f008:**
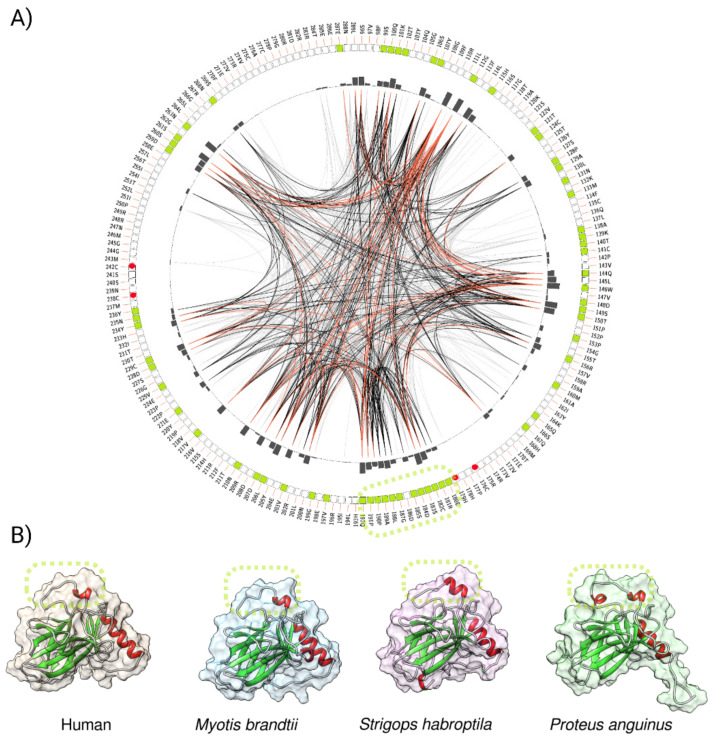
**A graphic representation of the positions of the p53 amino acids’ linked to longevity in the animal kingdom.** (**A**) Mutual information to infer the convergent evolution of p53 core domains. A circos plot is a sequential circular representation of the multiple sequence alignment and the information it contains. Green boxes in the outer circle indicate the positions of the amino acids’ changes correlating with longevity. The dashed oval highlights the longest region associated with longevity, which spans the loop 2 (L2, residues 180–192) region of human p53 DBD including S185. Lines connect pairs of positions with mutual information greater than 6.5 [51]. Red edges represent the top 5%, black represents between 70 and 95%, and gray edges account for the remaining 70%. (**B**) p53 core domains of three different, long-lived organisms compared to humans, as modeled by trRosetta.

**Figure 9 ijms-22-08512-f009:**
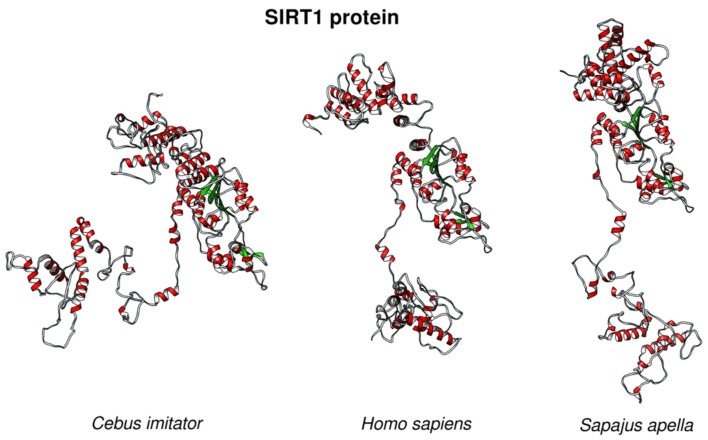
**The 3D structures of SIRT1 proteins from long-lived species compared to *Homo sapiens***. SIRT1 structures from *Cebus imitator* (XP_017357564.1), *Homo sapiens* (NP_001135970.1), and *Sapajus apella* SIRT1 (XP_032108492.1) showed differences in the protein structures in the given species.

**Figure 10 ijms-22-08512-f010:**
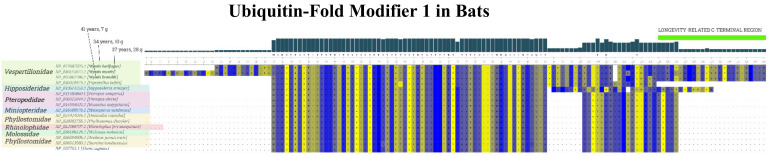
**Sequence alignment of UFM1 proteins in bat species and humans.** The 20-amino-acid-long extension of the C-terminal end in three long-living bats is depicted in light green. Multiple sequence protein alignments of UFM1 reference protein sequences were performed in MUSCLE with default parameters [47]; the colors express strand propensity.

**Figure 11 ijms-22-08512-f011:**
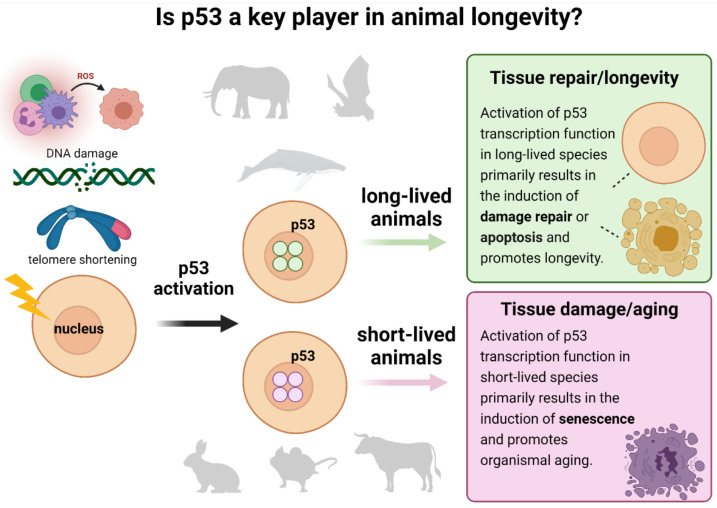
**Proposed p53-centric theory of extreme longevity.** Cell damage caused by ROS, DNA damage, telomere shortening, or other factors activates p53 to enable DNA repair and/or apoptosis. On the other hand, a high activity of p53 promotes organismal aging, thus shortening the lifespan. We hypothesize that long-lived animals developed the “improved” p53 proteins, which are less active than in their short-lived counterparts but still may sufficiently contribute to DNA damage repair and apoptosis in species that are exposed to environmental genotoxic stresses.

**Table 1 ijms-22-08512-t001:** Comparison of animals that were characterized by extreme longevity and their atypical p53 features, where the significance of particular changes was predicted. The default PROVEAN threshold of −2.5 was used, insertions and deletions were submitted relative to the human canonical protein sequence (NP_001119584.1). “*” indicates significant PROVEAN values (<−2.5).

Organism Classification	Maximal Lifespan (y)	Adult Weight (kg)	p53 Oddities	Effect Predicted by PROVEAN
*Balaena mysticetus* Mammalia, Cetacea	211	100,000	Unique substitution in proline rich region	Neutral (P77L, score = −0.993)
*Myotis brandtii, Myotis lucifugus* Mammalia, Chiroptera	41	0.007	Insertion in DNA-binding domain	Deleterious (P295_H296insPKQPPGS, score = −2.526) *
*Strigops habroptila* Aves, Psittaciformes	95	1.75	Substitution in core domain	Deleterious (N131H, score = −3.162) *
*Proteus anguinus* Amphibia, Urodela	102	0.017	Insertion nearby dimerization region	Deleterious (L188_A189insSR, score = −3.357) *
*Turritopsis* sp. Cnidaria	∞ rejuvenation	0.001	No p53/63/73 protein expressed (unprecedented phenomenon in the whole animal kingdom)	Not applicable

## Data Availability

The data presented in this study are available in Appendix A.

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
