# Peer review of "The Changes in the p53 Protein across the Animal Kingdom Point to Its Involvement in Longevity"

_ijms, 2021, doi:10.3390/ijms22168512_

Round 1
Reviewer 1 Report
This manuscript reports a comparative evolution study of the tumor suppressor p53 protein focusing on p53 sequence diversity and its correlation with lifespan. However, the biological process of aging and how it acts to control lifespan is a complex topic and the underlying mechanism still remains unclear. Although p53 has emerged as a player in the regulation of aging and longevity in worms, flies, mice, and humans, its role is complex and context dependent. As p53 is a major transcriptional factor and involved in the regulation of multiple signaling pathway, in some contexts, p53 can inhibit aging process and in other contexts, it can accelerate the process. Therefore, the reported comparative evolution study may provide some insight to p53 gene evolution and its impact on the protein activity in different organisms. However, the correlation of p53 diversity with lifespan of different organisms is not supported by the available evidence.
- Longevity is multifactorial topic and influenced by environmental risks and genetic factors. There are many genes that have been reported to be associated with longevity and aging. Using a correlation study simplify comparing the lifespan and diversity of p53 protein sequence would not be a valid scientific approach to address the topic.
- The structural and functional prediction provides a good basis for the functional studies of p53. The diversity of the protein sequence may reflect the the functional diversity of p53 in different organisms. However, there is not enough evidence to support the hypothesis that p53 protein sequence diversity is correlated with lifespan.
- As for figure 7, authors provided conflict data in the sequence specific association of p53 DNA-binding domain with the maximal lifespan in years. (a) The presented amino acids that appear to be associated with maximal lifespan are very different in different taxonomy categories, and there is no consensus in specific sets of amino acids that showed significant correlation in all organisms. (b) Using the residue 148D of p53 core domain as an example, the substitution of S appeared to be negatively associated with maximal lifespan in Glires, Chiroptera, Carnivora, but showed a positive association with maximal life span in Artiodactyla. So the correlation of p53 diversity with the lifespan of different organisms is not supported by the data presented.
Author Response
Comments and Suggestions for Authors
This manuscript reports a comparative evolution study of the tumor suppressor p53 protein focusing on p53 sequence diversity and its correlation with lifespan. However, the biological process of aging and how it acts to control lifespan is a complex topic and the underlying mechanism still remains unclear. Although p53 has emerged as a player in the regulation of aging and longevity in worms, flies, mice, and humans, its role is complex and context dependent. As p53 is a major transcriptional factor and involved in the regulation of multiple signaling pathway, in some contexts, p53 can inhibit aging process and in other contexts, it can accelerate the process. Therefore, the reported comparative evolution study may provide some insight to p53 gene evolution and its impact on the protein activity in different organisms. However, the correlation of p53 diversity with lifespan of different organisms is not supported by the available evidence.
Thank you for your comments. Aging is a complex molecular and biological process, nonetheless, the key role of p53 protein in aging becomes more and more evident, as we have described in the Introduction section. We agree, that the role of p53 in aging is complex and can be context-dependent. In our study, using available tools, we have demonstrated that animals with extremely long lifespans have certain p53 variations, which correlate with the organismal longevity. This was statistically tested and extensively discussed from several angles.
In addition, we have now discussed thoroughly our findings showing that the region of p53, spanning the residues of loop 2 in human p53, is associated with longevity (Figure 8a and b). We speculate that changes in p53 L2 in long-lived species are needed to alter the p53-driven cellular senescence. We have added the following text to our manuscript.
Page 12, line 369
The longest region in p53 associated with longevity spans amino acids in loop 2 (L2) of human p53 (Figure 8a, green, dashed circle). L2 is the minor groove binding region of human p53 and the stability of this region is maintained by Zn2+. Loss of Zn2+ triggers aggregation of L2 and loss of the DNA binding specificity. The 3D structures of DBD in selected long-lived species reveal intrinsic variations in L2 structure when compared to human p53 (Figure 8b). It is possible that intrinsic changes in L2, due to amino acid change, in long-lived species, amend p53 DNA binding specificity and/or binding of co-factors via an allosteric mechanism and alter the p53-driven senescence.
Page 16, line 488
We have identified 180 – 192 region, corresponding to loop 2 (L2) region of human p53, as the most conserved in long-lived species. The 3D model revealed variations in L2 structure in long-lived species when compared to human p53. Loop 2 is responsible for binding to the minor groove and its structure depends on the presence of Zn2+. We speculate that in long-lived species, L2 affects the p53 binding to DNA and/or other transcription factors and consequently, affects senescence program. On other hand, in humans, the L1 region is responsible for p53 binding to the major groove and was reported to undergo the most dynamic changes among DNA contacting loops (L1-L3) when located on non-target DNA sequence or on target DNA sequence [64]. Our findings indicate that L2 region, but not L1, might be needed to modulate the senescence (or other pro-aging) program in long-lived species, yet, detailed functional studies are needed to fully comprehend the role of p53 alterations in longevity.
Longevity is multifactorial topic and influenced by environmental risks and genetic factors. There are many genes that have been reported to be associated with longevity and aging. Using a correlation study simplify comparing the lifespan and diversity of p53 protein sequence would not be a valid scientific approach to address the topic.
Thank you for your comment. The current study is focused on p53 protein, since it is a critical sensor of cellular stresses including aging-related telomere shortening. We would like to emphasize that the total absence of p53 in the transcriptome of immortal jellyfish (Turritopsis sp.) is unprecedented and should be further inspected. Last, but not least, nearly all species with the changed p53 sequences are nocturnal animals, or animals living in absolute darkness (Strigops habroptila, Myotis brandtii, Myotis lucifugus, Proteus anguinus, Balaena mysticetus, Heterocephalus glabe), thus the low exposure to UV light was could have probably be a factor driving the gradual decrease in the activity of their p53s to evolve extreme longevity traits.
The structural and functional prediction provides a good basis for the functional studies of p53. The diversity of the protein sequence may reflect the the functional diversity of p53 in different organisms. However, there is not enough evidence to support the hypothesis that p53 protein sequence diversity is correlated with lifespan.
Thank you for the comment. Our structural and functional predictions provide good basis for further functional studies of p53 and allowed to uncover the region in p53 most significantly associated with longevity in the animal kingdom (residues 180-192 in DBD). In the previous version of the manuscript, we have not discussed thoroughly our findings on the putative role of the amino acid changes in loop 2 of human p53 in long-lived species (Figure 8a, b). We have now added additional description in Results and Discussion section as indicated above. At the same time, we are aware, that our work isn’t definitive, and many questions remain open (as discussed extensively in the Discussion section). We trust that our findings will lead to stimulation of further research in this field. In the future, we will perform the through functional studies on selected p53s identified in this work, yet, due to the COVID-19 pandemic, it is currently outside our capacity. All in all, we believe that our findings are well within the scope of the Journal and particularly of the Special Issue titled ‘Impacts of Molecular Structure on Nucleic Acid–Protein Interactions.’
As for figure 7, authors provided conflict data in the sequence specific association of p53 DNA-binding domain with the maximal lifespan in years. (a) The presented amino acids that appear to be associated with maximal lifespan are very different in different taxonomy categories, and there is no consensus in specific sets of amino acids that showed significant correlation in all organisms. (b) Using the residue 148D of p53 core domain as an example, the substitution of S appeared to be negatively associated with maximal lifespan in Glires, Chiroptera, Carnivora, but showed a positive association with maximal life span in Artiodactyla. So the correlation of p53 diversity with the lifespan of different organisms is not supported by the data presented.
Thank you for this comment. Yet, we would like to turn your attention to the fact that our analyses were performed for different subgroups of animals as we have not limited ourselves to long-lived species solely. The species included in our analysis have different evolutionary histories and different selection pressures within taxa and thus, 148S can be pro-longevity in Glires, Chiroptera, Carnivora, and pro-aging in Artiodactylia. Yet, we would like to emphasize again that the major conclusion from our study is that the region corresponding to human 180 – 192 aa (loop 2) is strongly associated with longevity across the animal kingdom. Further studies will answer the question how the changes in this region affect the binding of various p53s to target DNA sequence and/or transcriptional co-factors and how they affect the aging programs in long-lived species.
Reviewer 2 Report
The manuscript by Bartas et al described extensive analyses of p53 sequences across different animal species in an attempt to identify correlations between p53 variations with maximum lifespan. Previous work in mice and other model organisms suggested a role of hyperactive p53 in causing premature cellular and organismal aging, resulting in shortened lifespan. Bartas et al in this work revealed that organisms with extremely long lifespans tend to have p53 sequence variations such as amino acid substitutions and insertions in the core domain that are predicted to reduce DNA binding activity or alter target specificity. Furthermore, certain long-lived animals appear to have no p53 mRNA expression based on database search, suggesting they may have very low or no p53 activity. The findings suggest that p53 variants with reduced or altered DNA binding activity play a role in the long lifespan of certain animal species.
Overall, the manuscript is well-written, with nice summary of p53 background and its reported involvement in longevity. The results provide further support of the role of p53 in affecting certain animal natural lifespan. More importantly, the findings provide testable hypotheses for future experiments. A weakness of the manuscript is that functional prediction of the sequence variants is based on the impact of such changes if they are introduced into human p53 sequence, which there are data available from comprehensive mutation library analyses. It is possible that in the context of the unique p53 sequences of these distantly related species, the variations identified by the authors may not cause functional impairment or even have opposite effect. Therefore, the conclusions of the paper are intriguing but highly speculative in the complete absence of experimental data. Therefore, this is an interesting story but could be further strengthened with some experimental data to test the effects of some of the sequence changes, which should be within the capability of this research group.
Author Response
Comments and Suggestions for Authors
The manuscript by Bartas et al described extensive analyses of p53 sequences across different animal species in an attempt to identify correlations between p53 variations with maximum lifespan. Previous work in mice and other model organisms suggested a role of hyperactive p53 in causing premature cellular and organismal aging, resulting in shortened lifespan. Bartas et al in this work revealed that organisms with extremely long lifespans tend to have p53 sequence variations such as amino acid substitutions and insertions in the core domain that are predicted to reduce DNA binding activity or alter target specificity. Furthermore, certain long-lived animals appear to have no p53 mRNA expression based on database search, suggesting they may have very low or no p53 activity. The findings suggest that p53 variants with reduced or altered DNA binding activity play a role in the long lifespan of certain animal species.
Overall, the manuscript is well-written, with nice summary of p53 background and its reported involvement in longevity.
The results provide further support of the role of p53 in affecting certain animal natural lifespan. More importantly, the findings provide testable hypotheses for future experiments.
A weakness of the manuscript is that functional prediction of the sequence variants is based on the impact of such changes if they are introduced into human p53 sequence, which there are data available from comprehensive mutation library analyses. It is possible that in the context of the unique p53 sequences of these distantly related species, the variations identified by the authors may not cause functional impairment or even have opposite effect. Therefore, the conclusions of the paper are intriguing but highly speculative in the complete absence of experimental data. Therefore, this is an interesting story but could be further strengthened with some experimental data to test the effects of some of the sequence changes, which should be within the capability of this research group.
We would like to thank the Reviewer for the valuable feedback. To understand the impact of the changes in the p53 protein in different species, we have used PROVEAN tool and performed modelling of selected p53 DNA binding domains using SWISS-MODEL. We agree that these tools do not allow accurate prediction of p53 functionality in the studied species, yet no better tools are available to date. Functional studies are needed to fully comprehend the activity of different p53 proteins in the examined species, which is outside the scope of our current work and will be addressed in the follow-up study.
We have now emphasized the involvement of the loop 2 amino acids (corresponding to human p53 region 180-192) in longevity and added the following text in the Results and Discussion sections:
Page 12, line 369
The longest region in p53 associated with longevity spans amino acids in loop 2 (L2) of human p53 (Figure 8a, green, dashed circle). L2 is the minor groove binding region of human p53 and the stability of this region is maintained by Zn2+. Loss of Zn2+ triggers aggregation of L2 and loss of the DNA binding specificity. The 3D structures of DBD in selected long-lived species reveal intrinsic variations in L2 structure when compared to human p53 (Figure 8b). It is possible that intrinsic changes in L2, due to amino acid change, in long-lived species, amend p53 DNA binding specificity and/or binding of co-factors via an allosteric mechanism and alter the p53-driven senescence.
Page 16, line 488
We have identified 180 – 192 region, corresponding to loop 2 (L2) region of human p53, as the longest region associated with longevity. The 3D model revealed variations in L2 structure in long-lived species when compared to human p53. Loop 2 is responsible for binding to the minor groove and its structure depends on the presence of Zn2+. We speculate that in long-lived species, L2 affects the p53 binding to DNA and/or other transcription factors and consequently, affects senescence program. On other hand, in humans, the L1 region is responsible for p53 binding to the major groove and was reported to undergo the most dynamic changes among DNA contacting loops (L1-L3) when located on non-target DNA sequence or on target DNA sequence [64]. Our findings indicate that L2 region, but not L1, might play a role in modulating the senescence (or other pro-aging) program in long-lived species. Yet, detailed functional studies are needed to fully comprehend the role of p53 alterations in longevity.
We have also corrected minor language errors through the text.